# Incidence and Transition of Acute Kidney Injury, Acute Kidney Disease to Chronic Kidney Disease after Acute Type A Aortic Dissection Surgery

**DOI:** 10.3390/jcm10204769

**Published:** 2021-10-18

**Authors:** Chih-Hsiang Chang, Shao-Wei Chen, Jia-Jin Chen, Yi-Hsin Chan, Chieh-Li Yen, Tao Han Lee, Yu-Ting Cheng

**Affiliations:** 1Department of Nephrology, Chang Gung Memorial Hospital, Linkou Main Branch, Taoyuan City 33305, Taiwan; franwisandsun@gmail.com (C.-H.C.); b9102087@yahoo.com.tw (C.-L.Y.); kate0327@hotmail.com (T.H.L.); 2Kidney Research Center, Department of Nephrology, Chang Gung Memorial Hospital, Taoyuan City 33305, Taiwan; 3Division of Thoracic and Cardiovascular Surgery, Department of Surgery, Chang Gung Memorial Hospital, Linkou Medical Center, Taoyuan City 33305, Taiwan; Cooltony@gmail.com; 4Center for Big Data Analytics and Statistics, Chang Gung Memorial Hospital, Linkou Medical Center, Taoyuan City 33305, Taiwan; 5Department of Cardiology, Chang Gung Memorial Hospital, Linkou Medical Center, Taoyuan City 33305, Taiwan; s851047@hotmail.com

**Keywords:** acute kidney disease, acute kidney injury, aortic dissection, cardiovascular surgery

## Abstract

Acute kidney disease (AKD) is the persistence of renal injury between days 8 and 90 after an initial acute kidney injury (AKI). In this study, we aimed to explore the incidence of AKD, the association between AKD, and patient outcomes after acute type A aortic dissection (type A AAD) surgery. We identified 696 participants who underwent type A AAD surgery. Patients were categorized into stages 1 to 3 or 0 (non-AKD) AKD groups. Outcomes included major adverse kidney events (MAKEs), respiratory failure, all-cause readmission, and ischemic stroke from day 91 after operation. A total of 376 (54%) participants developed AKI, and 135/376 (35.9%) developed AKD. Moreover, 34/320 (10.6%) patients without AKI still developed AKD. Overall, 169/696 (24.3%) participants developed AKD. Patients with stages 2 and 3 AKD are associated with persisted declined renal function within 1 year. AKD was associated with a higher risk of MAKEs (hazard ratio (HR): 2.52, 95% confidence interval (CI) 1.90–3.33) and all-cause readmission (HR: 2.86, 95% CI: 2.10–3.89). Development of AKD with or without AKI is associated with a higher risk of MAKEs and hospitalization after acute aortic dissection surgery. Higher-stage AKD is associated with a trend of persistent decline in kidney function.

## 1. Introduction

Acute type A aortic dissection (type A AAD) is a medical emergency with high in-hospital mortality ranging from 22 to 31% [1,2,3]. Postoperative acute kidney injury (AKI) is a common complication with a high odds ratio of 3.49 for 30-day mortality after operation [4]. The incidence rate of AKI after cardiac surgery varies depending on population type and AKI criteria with the incidence rate ranging from 2 to 50% [5,6,7,8].

AKI is defined as kidney injury within 7 d after insult. Chronic kidney injury refers to kidney damage or dysfunction lasting more than 3 months. Acute kidney disease (AKD) is a transition between these two conceptualized models. AKD is a term used to describe the course of sustained kidney damage or kidney function nonrecovery that persists for more than 7 d but within 90 d after initial AKI [9]. One study demonstrated that AKD after cardiac surgery was associated with 90-day mortality and renal function decline [10].

However, the incidence and trajectory of renal function 8–90 d after operation, as well as the long-term prognostic effect of AKD, are poorly studied. We studied the incidence of AKD after type A AAD surgery to evaluate the transition between AKI and AKD and the trajectory of kidney function after postoperative AKD. Moreover, we evaluated the prognostic effect of AKD post-operation.

## 2. Materials and Methods

### 2.1. Data Source

This was a retrospective cohort study. The Chang Gung Research Database (CGRD) was used, which is an electronic medical record (EMR) database for 6 branches of Chang Gung Memorial Hospital (CGMH), from southern to northern Taiwan. In 2015, the CGMH provided health care to approximately 12% of the population under Taiwan’s National Health Insurance (NHI) program. The CGRD contains detailed data on diagnoses, prescriptions, vital signs, laboratory tests, and physical examination records from the emergency department and inpatient and outpatient settings. Renal diseases were defined using the International Classification of Diseases, Ninth Revision, Clinical Modification (ICD-9-CM) diagnostic codes before 2015 and the Tenth Revision (ICD-10-CM) diagnostic codes after 2016. More information regarding the CGRD can be obtained from the literature [11,12]. This study was approved by Chang Gung Medical Foundation Institutional Review Board, IRB number: 01801470B0. The identification number and personal information of the patients were encrypted and deidentified; therefore, informed consent was waived by Chang Gung Medical Foundation Institutional Review Board.

### 2.2. Study Population

Patients who were admitted and had an operation for acute type A AAD between 1 January 2004 and 31 December 2018 were identified. AAD surgery was confirmed if 3 criteria were fulfilled, namely a diagnostic code of AAD was used (ICD-9-CM: 441.0 and ICD-10-CM: I71.0), Taiwan’s NHI reimbursement codes for type A AAD surgery were identified, and a diagnosis of type A AAD surgery was made after 2 cardiovascular surgeons (Y.T. Cheng and S.W. Chen) reviewed the operation notes. A total of 44 patients received repeated type A AAD surgery during the study, and the first surgery was chosen as the index surgery. Patients were excluded if they were younger than 20 years old, were diagnosed as having end-stage renal disease and were on maintenance dialysis, received dialysis before the operation, died within 7 d after the operation, had insufficient creatinine data to enable a determination of whether they had AKI or AKD, and had a follow-up of less than 90 d (including death between 8 d and 90 d) (Figure 1).

### 2.3. Definition of AKI and AKD

The exposure variables were AKI or AKD after type A AAD. AKI was defined based on the criteria of Kidney Disease Improving Global Outcomes (KDIGO) [13]. The highest creatinine levels within 7 d after an operation were compared with baseline creatinine levels for AKI diagnosis and stage classification. Baseline creatinine was defined as the value recorded upon emergency room arrival or on the day of admission. AKD was defined according to the Acute Disease Quality Initiative (ADQI) 16 Workgroup consensus [9]. AKD stages 1, 2, and 3 were identified as serum creatinine levels 1.5 to 1.9 times baseline creatinine; 2.0 to 2.9 times baseline creatinine; at least 3.0 times baseline creatinine, a creatinine level of ≥4.0 mg/dL, or renal placement therapy 8 to 90 d after the index date, respectively [9]. If more than 1 value of creatinine level was obtained 8 to 90 d after the index surgery, AKD stage was defined using creatinine levels measured closest to day 90.

### 2.4. Covariates

Covariates included patient demographics (age, gender, body weight, height, body mass index (BMI), and smoking habit), comorbidities (Marfan syndrome, diabetes mellitus, hypertension, dyslipidemia, chronic kidney disease (CKD), previous stroke, coronary artery disease, chronic obstructive pulmonary disease, atrial fibrillation, and peripheral arterial disease), preoperative vital signs (mean arterial pressure and heart rate), and preoperative laboratory examination data (creatinine, blood urea nitrogen (BUN), white blood cell (WBC) count, hemoglobin (Hb), platelet count, and sodium and potassium levels). Surgical details were obtained through a review of operation notes by 2 cardiovascular surgeons (Y.T. Cheng and S.W. Chen), including bypass time, arrest time, method for brain protection, cerebral perfusion time, and extension for type A surgery. Postoperative sequential organ failure assessment (SOFA) scores and postoperative laboratory results, including aspartate aminotransferase (AST), alanine aminotransferase (ALT), hemoglobulin, and platelet count, were extracted. Comorbidity was defined based on at least 2 outpatient diagnoses or an inpatient diagnosis before admission for an index surgery (Appendix A).

### 2.5. Outcomes

AKD was evaluated from day 8 to day 90 after the index surgery, and therefore, late outcomes were analyzed from day 91 after the index surgery. The primary late outcome was a composite of major adverse kidney events (MAKEs), including recurrent AKI, de novo CKD (estimated glomerular filtration rate of <60 mL/min/1.73 m^2^, according to the Modification of Diet in Renal Disease (MDRD) equation for non-chronic kidney disease patients), end-stage renal disease (ESRD), and all-cause mortality from day 91 of the index surgery to the end of follow-up. Newly diagnosed CKD or de novo CKD was only identified in those with baseline eGFR more than 60 mL/min/1.73 m^2^. Recurrent AKI was defined as an increase of creatinine by 0.3 mg/dL or with increased creatinine 1.5 times within any 48 h from 91 d after index surgery to the end of follow-up. Moreover, secondary outcomes, including all-cause readmission, respiratory failure, and ischemic stroke, were analyzed. The incidence of respiratory failure and ischemic stroke was identified using the ICD-9-CM and ICD-10-CM diagnostic codes from the inpatient claims data (Appendix A). Patients were followed until the date of event occurrence, the date of death, or 31 December 2018, whichever came first.

### 2.6. Statistical Analysis

The baseline and clinical characteristics of patients in the study groups (i.e., AKI vs. non-AKI; AKD vs. non-AKD) were compared using the chi-squared test for categorical variables, independent sample *t* test for continuous variables with a normal distribution, and Mann–Whitney U test for continuous variables without a normal distribution. The association between AKD incidence and the risk of late outcomes was investigated using univariate and multivariate Cox proportional hazards models. Clinically relevant covariates, including AKI stage, age, gender, BMI, smoking habit, diabetes mellitus, hypertension, chronic kidney disease, coronary artery disease, mean arterial pressure, preoperative creatinine and hemoglobin levels, bypass time, shock in the operating room, and extensions for type A surgery, were adjusted in the Cox model. As ESRD (dialysis) was a competing risk for recurrent AKI and newly diagnosed CKD, we conducted the Fine and Gray subdistribution hazard model, which considered ESRD a competing risk. Due to the substantial number of missing values, the survival analyses were performed using the single imputation expectation–maximization cohort. Additionally, the assumption of proportional hazard was assessed using the Schoenfeld residuals method on the primary renal-related outcomes. A two-sided *p*-value of <0.05 was considered to represent statistical significance. All statistical analyses were performed using SAS version 9.4 (SAS Institute, Cary, NC, USA). The directly adjusted survival rates of the AKD and non-AKD groups were calculated using the SAS macro “%ADJSURV” [14].

## 3. Results

### 3.1. Patient Characteristics

The mean age of the patients was 57.6 years (standard deviation: 13.7 years), and the male sex was predominant (67.8%). The average BMI value was 25.9 kg/m^2^ (25th to 75th percentile: 23.3 to 29.4 kg/m^2^). Approximately, 40% of the patients smoked. Most patients (71.8%; 500/696) had hypertension and 10.9% (76/696) were diagnosed as having CKD. The median hospital stay was 22 d (Appendix A). Compared to patients without AKI, patients with AKI were more likely to be men and had higher BMIs; a higher prevalence of baseline CKD; higher levels of preoperative creatinine, BUN, Hb, and sodium; higher WBC counts; lower preoperative platelet counts; longer surgical times; higher postoperative SOFA scores; higher levels of postoperative AST and ALT; lower postoperative Hb levels and platelet counts; and longer stays at the hospital. Higher bypass times, higher clamp times, and arrest times were also found in the AKI group in our cohort. Compared to patients without AKD, patients with AKD had higher BMIs; a lower prevalence of hypertension; a higher prevalence of CKD; higher levels of preoperative creatinine and BUN; longer surgery times; higher postoperative SOFA scores; higher levels of postoperative AST; lower postoperative platelet counts; longer hospital stays; and a higher incidence of AKI (Appendix A). We also conducted a multivariable logistic regression analysis to analyze the association of covariates with AKI and AKD. The results are summarized in Appendix A.

### 3.2. Incidence of AKD after Surgery

After the exclusion criteria were applied, 696 patients were eligible for analysis, of which 376 (54%) developed AKI. Furthermore, of the 376 patients with AKI, 135 (35.9%) progressed to AKD. Notably, among the 320 patients without AKI, 34 (10.6%) had deteriorated kidney function, which progressed to AKD. Overall, 169 of 696 (24.3%) patients developed AKD after type A AAD surgery (Figure 1).

### 3.3. Trajectory of Renal Function

After operations in patients with AKI or AKD, creatinine levels increased gradually and peaked 2 to 3 d after operations in patients with AKI (Figure 2A). The creatinine levels of patients with stage 1 or 2 AKI peaked within 2 d after their operations, then gradually decreased to baseline around 7 d after their operations. Baseline creatinine levels in patients with stage 3 AKI were higher than in patients with stage 1 or 2 AKI. The creatinine levels of patients with stage 3 AKI peaked within 3 d after their operations, then gradually decreased (Figure 2A). If a patient with severe AKI required dialysis, creatinine levels after dialysis were not included in this analysis (Figure 2A).

Renal function persistently declined in patients with stage 2 or 3 AKD within 90 d after their operations, and the trend persisted for up to 1 year after their operations compared to patients with stage 0 or 1 AKD. The average estimated glomerular filtration rate (eGFR) of patients with stage 2 or 3 AKD gradually decreased with a nadir eGFR < 60 around 180 d after their operations. Renal function in patients with stage 0 AKD or stage 1 AKI was relatively stable 91 to 365 d after their operations (Figure 2B). Creatinine levels after dialysis were not evaluated (Figure 2B).

### 3.4. Association between AKD and the Risk of Late Outcomes

The overall mean follow-up was 4.4 years (standard deviation: 3.4 years). The event rate of MAKEs was 65.7% (111/169) in the AKD group and 42.1% (222/527) in the non-AKD group (Table 1). After multivariate adjustments were made, AKD was significantly associated with a higher risk of MAKEs in comparison to the non-AKD group (hazard ratio (HR) 2.52, 95% confidence interval (CI) 1.90–3.33) (Figure 3A). Notably, only newly diagnosed CKD was significant (55% vs. 38.1%; HR 2.86, 95% CI 2.10–3.89) among the MAKEs (Figure 3B). The AKD group had a significantly higher risk of all-cause readmission in comparison to the non-AKD group (59.8% vs. 44.8%; HR 1.45, 95% CI 1.09–1.92). The AKD group showed a trend of higher risks of respiratory failure and ischemic stroke but without statistical significance. The results of recurrent AKI and newly diagnosed CKD remained consistent in the competing risk survival analysis (Appendix A). In addition, the correlation of the Schoenfeld residuals was −0.12, −0.14, and −0.15 for MAKE, recurrent AKI, and newly diagnosed CKD, respectively. A result was not available for ESRD because of the limited events. These low correlation coefficients suggested there were no apparent violations of proportional hazard assumption.

## 4. Discussion

AKD is a subacute, persistent kidney injury or kidney function nonrecovery between 8 and 90 d after initial AKI. Studies have revealed that the timing of renal function recovery or persistence of AKI might result in different outcomes [9,15]. The trajectory from AKI to AKD and the long-term outcomes of AKD after type A AAD are poorly understood. The four major findings of our study can be summarized as follows: (1) the overall incidence of stages 1 to 3 AKD after operation was 24.3%; (2) the incidence of stages 1 to 3 AKD in patients with AKI after operation was 35.9%; moreover, 10.6% of those without an AKI episode developed stage 1, 2, or 3 AKD; (3) patients with stage 2 or 3 AKD had a gradual decline of renal function during the 1-year follow-up; (4) the development of AKD was associated with a higher risk of MAKEs and all-cause readmission.

The incidence of AKD after type A AAD had previously not been reported. Matsuura et al. reported that 11.2% of patients developed stages 1, 2, or 3 AKD after cardiac surgery [10]. The incidence of AKD in patients without an AKI episode varied from 17 to 24% [16,17]. Possible explanations for the diagnosis of stages 1 to 3 AKD with no AKI episode included subclinical AKI, recurrent AKI after initial subclinical AKI, and drug or toxin exposure [9]. The overall incidence of stages 1 to 3 AKD in our study was 24.3%. Moreover, 35.9% of patients with postoperative AKI and 10.6% of patients without AKI developed stage 1, 2, or 3 AKD. Through the multivariable logistic regression analysis, we also identified several risks for AKI, including chronic kidney disease, baseline renal function, pre-operation hemoglobin level, and bypass time (Appendix A). The identified risk factors of AKD were AKI development and the AKI severity/stage (Appendix A).

Matsuura et al. [10] reported that any stage of AKD is a risk factor for kidney function decline, with an odds ratio of 3.56 for a 50% eGFR decline in hospital survivors during the 2-year follow-up. Moreover, although 90-day mortality after the operation was found to be strongly associated with AKD, no long-term outcome was analyzed in their study [10]. We analyzed the long-term outcome of AKD and found that AKD was associated with a high risk of MAKEs. AKD was found to be associated with a trend of higher mortality, respiratory failure, and ischemic stroke; however, the difference was statistically insignificant. Compared with patients with stage 1 AKD, patients with stage 2 or 3 AKD had persistent renal function decline in 1 year. Any stage of AKD was a predictor of de novo CKD development (HR: 2.86). Thus, in patients with a higher stage of AKD, early referral to a nephrologist or timely intervention may be important, which is concordant with the ADQI consensus (9). Current knowledge about management of severe AKI and AKD is still suboptimal. One recent study showed the controversial role of early renal replacement therapy (RRT) initiation in the cardiac surgery population [18]. Furthermore, the presentation of post-AKI proteinuria is a poor prognostic indicator [19]. Trials have demonstrated the possible beneficial role of angiotensin-converting enzyme inhibitors/angiotensin receptor blockers (ACEI/ARB) in the AKI population, which might reduce mortality [20,21,22]. A study by Chen et. al. also revealed improved prognosis through the prescription of a β-blocker after aortic surgery [23]. Though the protective effect of ACEI/ARB was not detected in the AKIKI-2 study, which enrolled a post-surgical AKI population [24], a recent meta-analysis demonstrated potential survival using an ACEI/ARB prescription in the post-AKI population [25]. Therefore, studies on the prognostic effect of prescribed medications, including ACEI/ARB, in patients with post-cardiac surgery AKI and AKD are warranted.

Our study has several limitations. First, this study was retrospective. Second, owing to the lack of a biomarker for kidney damage in this cohort, we could not group patients with stage 0 AKD into subgroups, including stages 0A, 0B, and 0C. Third, some patients did not receive regular long-term follow-up, and the detailed trajectory of renal function, according to different AKD stages, could not be analyzed. Therefore, the renal function trajectory of a larger cohort of patients with AKD should be evaluated. Moreover, prospective studies assessing the role of early intervention in patients with AKD at a high risk of CKD are required. Fourth, though patients in our study cohort had long-term all-cause mortality similar to a previous nationally based study [23], it is still possible patients who had outcome events were not followed up in our hospital. Therefore, we might underestimate the outcome of interest. Fifth, since most patients (>80%) did not have baseline creatinine from 7 to 365 d before the index surgery, we used the first creatinine examination at the arrival at the emergency department or at the first day of the index admission as the baseline creatinine, which might underestimate community-acquired AKI, underestimate the incidence of recurrent AKI after discharge, and overestimate baseline CKD prevalence [26]. Furthermore, we did not conduct sample-size calculations, owing to the retrospective cohort study design. However, we did perform a post hoc power analysis of our outcomes with significant findings (including MAKEs, newly diagnosis CKD, all-cause readmissions), and the results showed the achieved power was 99.92%, 99.98%, and 54.4% for MAKEs, newly diagnosis CKD, and all-cause readmissions, respectively (data not shown).

## 5. Conclusions

AKD with or without AKI episodes after aortic dissection is associated with a higher incidence of MAKEs. A persistent decline in renal function was observed after the initial insult in patients with stage 2 or 3 AKD.

## Figures and Tables

**Figure 1 jcm-10-04769-f001:**
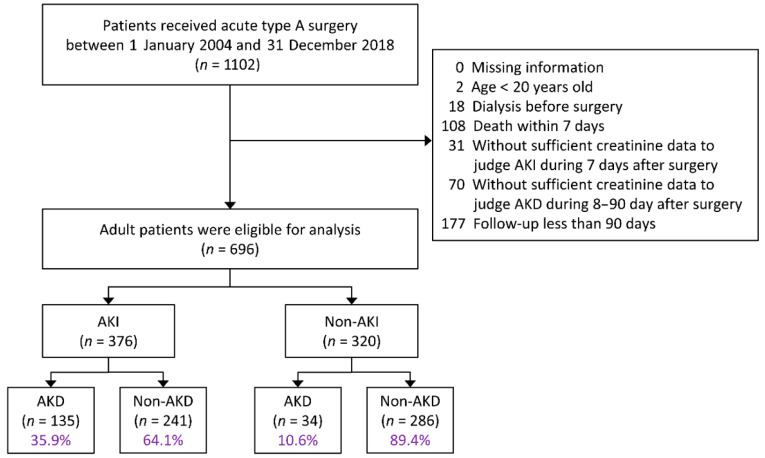
Diagram of patient selection. AKI, acute kidney injury; AKD, acute kidney disease.

**Figure 2 jcm-10-04769-f002:**
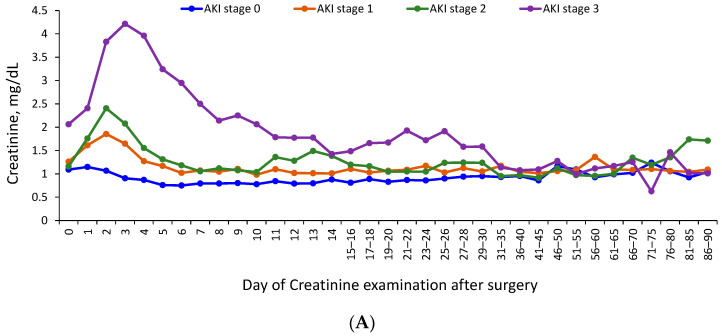
Kidney function trajectory after acute kidney injury and acute kidney disease. (**A**) Trend of creatinine levels after operation according to AKI stage. AKI, acute kidney injury. Creatinine or eGFR data after dialysis initiation were not included in the figure. AKD, acute kidney disease; eGFR, estimated glomerular filtration rate. (**B**) Trend of eGFR after operation according to AKD stage. Creatinine or eGFR data after dialysis initiation were not included in the figure. AKD, acute kidney disease; eGFR, estimated glomerular filtration rate.

**Figure 3 jcm-10-04769-f003:**
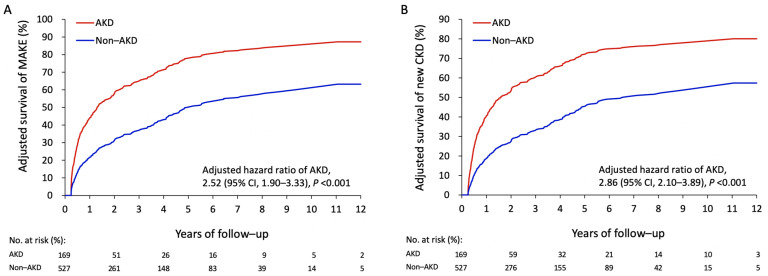
Directly adjusted one minus survival rates of MAKEs (**A**) and newly diagnosed CKD (**B**) after type A acute aortic dissection surgery in patients with or without AKD. MAKEs, major adverse kidney events; CKD, chronic kidney disease.

**Table 1 jcm-10-04769-t001:** Late outcomes during follow-up in the expectation maximization imputation cohort.

Late Outcomes (>90 Days)	AKD (*n* = 169)	Non-AKD (*n* = 527)	Unadjusted Analysis		Multivariable Analysis *	
			HR (95% CI) for AKD	*p*-Value	HR (95% CI) for AKD	*p*-Value
MAKEs	111 (65.7)	222 (42.1)	2.24 (1.76–2.86)	<0.001	2.52 (1.90–3.33)	<0.001
Recurrent AKI	35 (20.7)	77 (14.6)	1.57 (1.05–2.35)	0.029	1.24 (0.77–2.01)	0.372
Newly diagnosed CKD	93 (55.0)	201 (38.1)	1.95 (1.50–2.55)	<0.001	2.86 (2.10–3.89)	<0.001
ESRD	8 (4.7)	15 (2.8)	1.66 (0.72–3.85)	0.236	1.51 (0.65–3.48)	0.336
All-cause mortality	17 (10.1)	32 (6.1)	1.67 (0.92–3.03)	0.093	1.58 (0.81–3.09)	0.179
All-cause readmission	101 (59.8)	236 (44.8)	1.47 (1.16–1.86)	0.002	1.45 (1.09–1.92)	0.010
Respiratory failure	11 (6.5)	23 (4.4)	1.62 (0.79–3.33)	0.190	1.54 (0.61–3.90)	0.365
Ischemic stroke	14 (8.3)	31 (5.9)	1.39 (0.73–2.66)	0.322	1.10 (0.57–2.11)	0.786

Abbreviations: AKD, acute kidney disease; AKI, Acute kidney injury; HR, hazard ratio; CI, confidence interval; MAKEs, major adverse kidney events; CKD, chronic kidney disease; ESRD, end-stage renal disease; * Adjusted with AKI stage, age, gender, body mass index, smoking, diabetes mellitus, hypertension, chronic kidney disease, coronary artery disease, mean arterial pressure, pre-operation creatinine, pre-operation hemoglobin, bypass time, shock at operation room, extensions for type A surgery.

## Data Availability

The data presented in this study are available on request from the corresponding author. The data are not publicly available because the study is based in part on data from the Chang Gung Research Database provided by Chang Gung Memorial Hospital.

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
