# Peer review of "Incidence and Transition of Acute Kidney Injury, Acute Kidney Disease to Chronic Kidney Disease after Acute Type A Aortic Dissection Surgery"

_jcm, 2021, doi:10.3390/jcm10204769_

Round 1

Reviewer 1 Report

Chang et al. evaluate the AKI and AKI to CKD after type A aortic dissection surgery. I have some suggestions for this study.

1. Please provide the number of the ethical permit document in the method section.

2. The outcomes definition of MAKEs includes all-cause mortality. However, it is estimated from day 91 of the index surgery to the end of follow-up. A clear explain to use this definition is suggested.

3. Subjects expired within 7 days after acute type A surgery were excluded. How to evaluate the association of the outcome if subjects died between 8-90 days?

4. This study is conducted using Chang Gung Research Database (CGRD). Does the national death registry confirm the death?

5. According to the diagram of patient selection, subjects with AKI may present AKD or non-AKD. Subjects without AKI will present non-AKD. According to the general concept, acute kidney disease (AKD) is the persistence of renal injury between days 8 and 90 after an initial acute kidney injury (AKI). Thus, how to explain subjects without AKI (n=320) but present AKD (n=34) in this study? The classification of these 34 patients will have a significant impact on the outcome.

Author Response

Reviewer 1

Comments and Suggestions for Authors

Chang et al. evaluate the AKI and AKI to CKD after type A aortic dissection surgery. I have some suggestions for this study.

Point 1. Please provide the number of the ethical permit document in the method section.

Reply 1: We appreciate your kind reminding us. We provided IRB number in the revised manuscript (Tracking change version page 2, line 64).

Point 2. The outcomes definition of MAKEs includes all-cause mortality. However, it is estimated from day 91 of the index surgery to the end of follow-up. A clear explain to use this definition is suggested.

Reply 2: We appreciate your valuable comment. The definition of AKD is renal non-recovery or persisted kidney injury from 8 to 90 days after initial kidney insult. Therefore, late outcomes were analyzed from day 91 after the index surgery. We added further explanation into section 2.5. (Tracking change version page 4, line 115)

Point 3. Subjects expired within 7 days after acute type A surgery were excluded. How to evaluate the association of the outcome if subjects died between 8-90 days?

Reply 3: We appreciate your valuable comment. The main exposure we used for outcome analysis was acute kidney injury. Population who expired within 7 days after index surgery (n=108), who were without sufficient creatinine to judge AKD within 8 to 90 days after surgery (n=70), and those died between 8-90 days after surgery or had follow up less than 90 days (n=177) were all excluded (Figure 1).

Point 4. This study is conducted using Chang Gung Research Database (CGRD). Does the national death registry confirm the death?

Reply 4: We appreciate your insightful comment. Those who lost follow up from our hospital and death in other hospital might not be evaluated in the current study. We added this important point into our limitation session. However, our hospital is a referral center and most patients underwent type A dissection surgery would continue to be followed up in our outpatient department. Besides, the all-cause mortality in our cohort was 10% which was similar to our previous study based on the national death registry (reference 1). (Tracking change version page 8, line 284)

Reference 1: Chen SW, Lin YS, Wu VC, Lin MS, Chou AH, Chu PH, Chen TH. Effect of β-blocker therapy on late outcomes after surgical repair of type A aortic dissection. J Thorac Cardiovasc Surg. 2020 May;159(5):1694-1703.e3. doi: 10.1016/j.jtcvs.2019.05.032. Epub 2019 May 31. PMID: 31262538.

Point 5. According to the diagram of patient selection, subjects with AKI may present AKD or non-AKD. Subjects without AKI will present non-AKD. According to the general concept, acute kidney disease (AKD) is the persistence of renal injury between days 8 and 90 after an initial acute kidney injury (AKI). Thus, how to explain subjects without AKI (n=320) but present AKD (n=34) in this study? The classification of these 34 patients will have a significant impact on the outcome.

Reply 5: We appreciate your valuable comment. According to the ADQI AKD definition, AKD is identified by “a condition in which AKI stage 1 or greater, as defined by KDIGO, is present ≥7 days after an AKI initiating event”. Nevertheless, the same ADQI consensus also mentioned about that “an AKI initiating event can usually be identified but is not required to diagnose AKD” (reference 2). Besides, according to recently published review article from ADQI XXII meeting, the definition of AKD change to “ Prolonged kidney dysfunction (in the presence or absence of AKI) that may occur before a patient meets the 90-day criteria needed for the diagnostic criteria of chronic kidney disease (CKD)” (reference 3).

Reference 2: Chawla LS, Bellomo R, Bihorac A, Goldstein SL, Siew ED, Bagshaw SM, Bittleman D, Cruz D, Endre Z, Fitzgerald RL, Forni L, Kane-Gill SL, Hoste E, Koyner J, Liu KD, Macedo E, Mehta R, Murray P, Nadim M, Ostermann M, Palevsky PM, Pannu N, Rosner M, Wald R, Zarbock A, Ronco C, Kellum JA; Acute Disease Quality Initiative Workgroup 16.. Acute kidney disease and renal recovery: consensus report of the Acute Disease Quality Initiative (ADQI) 16 Workgroup. Nat Rev Nephrol. 2017 Apr;13(4):241-257. doi: 10.1038/nrneph.2017.2. Epub 2017 Feb 27. PMID: 28239173.

Reference 3: Liu KD, Forni LG, Heung M, Wu VC, Kellum JA, Mehta RL, Ronco C, Kashani K, Rosner MH, Haase M, Koyner JL; Acute Disease Quality Initiative Investigators. Quality of Care for Acute Kidney Disease: Current Knowledge Gaps and Future Directions. Kidney Int Rep. 2020 Aug 6;5(10):1634-1642. doi: 10.1016/j.ekir.2020.07.031. PMID: 33102955; PMCID: PMC7569680.

Reviewer 2 Report

The authors conducted a retrospective study using multicenter database to evaluate the clinical course of acute kidney disease after acute aortic dissection surgery. They show the trajectory of renal function after surgery and the association with AKD and various outcomes including MAKE. The result is interesting but there are some critical points to modify or clarify.

  1. Baseline serum creatinine was defined as the value at the emergency room or at the admission. This would be inappropriate because serum creatinine at the admission could be already elevated and AKI had had incidence. Generally, serum creatinine within 1 year before hospitalization is defined as baseline serum creatinine. Besides, please clarify how many patients have CKD.

  1. Primary outcome includes recurrent AKI. How did you define baseline serum creatinine for identifying recurrent AKI? The value at the index admission could not be used because some patients have worsen kidney function than at the index hospitalization.

  1. How did you define newly-diagnosed CKD for CKD patients?

  1. In Table 1, how did you deal with competing risks when evaluating the association with AKD and recurrent AKI, newly diagnosed CKD, ESRD?

  1. How did you check that these hazard ratios were constant over time when using Cox models?

Author Response

Reviewer 2

Comments and Suggestions for Authors

The authors conducted a retrospective study using multicenter database to evaluate the clinical course of acute kidney disease after acute aortic dissection surgery. They show the trajectory of renal function after surgery and the association with AKD and various outcomes including MAKE. The result is interesting but there are some critical points to modify or clarify.

Point 1. Baseline serum creatinine was defined as the value at the emergency room or at the admission. This would be inappropriate because serum creatinine at the admission could be already elevated and AKI had had incidence. Generally, serum creatinine within 1 year before hospitalization is defined as baseline serum creatinine. Besides, please clarify how many patients have CKD.

Reply 1: We appreciate your insightful comment. There are several different baseline creatinine level definition in published AKI study (reference 4). In our cohort, owing to acute type A aortic dissection in a medical emergency and we are a referral center, only less than 20% of our enrolled participants have available baseline creatinine (7 to 365 days before the index surgery), we used the first creatinine examination at the arrival of emergent department or at the first day of the index admission as the baseline creatinine , which might underestimated community acquired AKI. We added this point into our limitation in revised manuscript. Again, thanks for your valuable comment. (Tracking change version page 8, line 287).

Reference 4: Thomas ME, Blaine C, Dawnay A, Devonald MA, Ftouh S, Laing C, Latchem S, Lewington A, Milford DV, Ostermann M. The definition of acute kidney injury and its use in practice. Kidney Int. 2015 Jan;87(1):62-73. doi: 10.1038/ki.2014.328. Epub 2014 Oct 15. PMID: 25317932.

Point 2. Primary outcome includes recurrent AKI. How did you define baseline serum creatinine for identifying recurrent AKI? The value at the index admission could not be used because some patients have worsen kidney function than at the index hospitalization.

Reply 2: We appreciate your insightful comment and kindly reminding. We added the definition of recurrent AKI into the “2.5 Outcomes” subsection of the Methods session. Recurrent AKI was defined as an increase of creatinine by 0.3 mg/dl or with increased creatinine 1.5 times within any 48 hours from 91 days after index surgery to the end of follow up. Furthermore, similar to the your above comment, we used the first available creatinine in our hospital as the baseline creatinine when determining recurrent AKI, which might result in underestimating the incidence of community acquired AKI, overestimating CKD prevalence and underestimating the incidence of recurrent AKI after discharge. We addressed this problem more clearly in the discussion session. (Tracking change version page 4, line 123; page 8, line 287)

Point 3. How did you define newly-diagnosed CKD for CKD patients?

Reply 3: We thanks your kindly reminding. We defined the new-diagnosis CKD as de novo chronic kidney disease (estimated glomerular filtration rate of <60 mL/min/1.73 m2, according to the Modification of Diet in Renal Disease (MDRD) equation for non–chronic kidney disease patients). Therefore, in those patients already with CKD (baseline eGFR < 60), there will be no new-diagnosed CKD event in the survival analysis. We clarified this problem in the “2.5 Outcomes” subseciton of the Methods section (Tracking change version page 4, line 121).

Point 4. In Table 1, how did you deal with competing risks when evaluating the association with AKD and recurrent AKI, newly diagnosed CKD, ESRD?

Reply 4: Thanks for your comment. We agree with your opinion. ESRD (dialysis) would be a competing risk for recurrent AKI and newly diagnosed CKD, therefore we conducted the Fine and Gray subdistribution hazard model which considered ESRD a competing risk. The results of recurrent AKI and newly diagnosed CKD remained consistent in the competing risk survival analysis (Supplementary Table 3). We have added the relevant content in the “2.6 Statistical Analysis” subsection of the Methods section and the “3.4 Association Between AKD and the Risk of Late Outcomes subsection of the Results section” (Tracking change version page 4, line 141; page 6, line 212).

Point 5. How did you check that these hazard ratios were constant over time when using Cox models?

Reply 5: Thanks for your comment. The assumption of proportional hazard was assessed using the Schoenfeld residuals method on the primary renal-related outcomes: MAKE, recurrent AKI, newly-diagnosed CKD and ESRD. The correlation of the Schoenfeld residuals was -0.12, -0.14 and -0.15 for MAKE, recurrent AKI and newly-diagnosed CKD, respectively. The result was not available for ESRD because of the few events. These low correlation coefficients suggested there were no apparent violation of proportional hazard assumption. We have addressed this issue in the “2.6 Statistical Analysis” subsection of the Methods section and the “3.4 Association Between AKD and the Risk of Late Outcomes subsection of the Results section” (Tracking change version page 4, line 141; page 6, line 212).

Reviewer 3 Report

Dear Editor and Authors,

Thank you for asking me to review this manuscript by Dr. Chang and his colleagues from the Chang Gung Memorial Hospital in Taipei, Taiwan titled “Incidence and Transition of Acute Kidney Injury, Acute Kidney Disease to Chronic Kidney Disease after Acute Type A Aortic Dis-section Surgery”.

This is an interesting study in which the authors investigate the transition of acute kidney injury (AKI) to acute kidney disease (AKD) to chronic disease (CRD) in patient undergoing type A aortic dissection repair. This is an interesting study to me as a cardiothoracic surgeon and one that evidence is lacking in the literature. There is a lot of work naturally on AKI but not so much on the renal sequelae following this type of cardiac surgery.

The article is well written and easily understood. It is a retrospective database review – analysis which of course does pose a limitation in terms of availability of data variables for analysis and completeness. It contains originally a large sample size of 696 patients however there is no power analysis or sample size calculation to confirm the study is adequately powered to produce meaningful statistical results! This is unfortunately an omission by the authors at the beginning stage of designing the study and although my feeling is that given the significant incidence of AKI and AKD the results are most likely accurate the authors have to now address this issue as a limitation of their study!! Can they ask their statistician to perform a post-hoc power analysis or to explain how the results are robust?

Why did they not perform a multivariable analysis to delineate better the associated co-variables for AKI and AKD. I am aware that there is a significant work done on that particular subject and variables that cause AKI and AKD post cardiac surgery are well known and reported but they need to do the analysis to demonstrate that their population cohort is comparable to others reported! They can report the outcomes of such analysis as a supplementary table as they have done with other data to avoid overburdening the reader with information and risking loosing the focus of the study which is on the correlation of AKI, AKD and long term CRD!

Why did the authors/statistician utilize the Cox proportional hazards model to investigate the association between AKD incidence to long term outcomes such as MAKEs, New CKD, ESRD ect and not propensity score matching which would have been more statistically robust?

According to supplementary table 2 patients which developed AKI also had longer cardiopulmonary bypass times, clamp times and arrest times. To a cardiothoracic surgeon this is natural and expected associated variables; however I did not see this mentioned in the results text and possibly it needs to be presented and addressed for the non-surgical audience of the paper!

The discussion is well presented, poignant and succinct however I would like to see some mention about surgical causes of AKI post cardiac surgery (mentioned about performing a multivariable analysis of associated co-variables). Also, the authors very well mention that medications that patients receive post dissection surgery to control their blood pressure can contribute both positive but also some negatively (the authors have not taken to mention this point in their discussion) in the development of long term CRD!

In conclusion, I like this study! It is a retrospective analysis but the concept/hypothesis is novel and uninvestigated. The authors have performed a good analysis which shows me that they are knowledgeable about conducting such studies, therefore I am surprised that the analysis is incomplete in certain aspects (i.e. lacking power analysis - sample size calculation, no multivariable analysis of associated factors, ect). I encourage the authors to perform these and modify their manuscript accordingly because, even though as I said they are not omissions that make the study rejectable, their inclusion will make it more robust and useful. I wish all well.  

Author Response

Reviewer 3

Comments and Suggestions for Authors

This is an interesting study in which the authors investigate the transition of acute kidney injury (AKI) to acute kidney disease (AKD) to chronic disease (CRD) in patient undergoing type A aortic dissection repair. This is an interesting study to me as a cardiothoracic surgeon and one that evidence is lacking in the literature. There is a lot of work naturally on AKI but not so much on the renal sequelae following this type of cardiac surgery.

Point 1. The article is well written and easily understood. It is a retrospective database review – analysis which of course does pose a limitation in terms of availability of data variables for analysis and completeness. It contains originally a large sample size of 696 patients however there is no power analysis or sample size calculation to confirm the study is adequately powered to produce meaningful statistical results! This is unfortunately an omission by the authors at the beginning stage of designing the study and although my feeling is that given the significant incidence of AKI and AKD the results are most likely accurate the authors have to now address this issue as a limitation of their study!! Can they ask their statistician to perform a post-hoc power analysis or to explain how the results are robust?

Reply 1: We appreciate your kind comment and reminding us. Due to retrospective cohort design, we did not calculate sample size in the beginning of the study. However, we addressed this important problem via post-hoc power analysis on the three outcomes with significant finding (including MAKE, new-diagnosis CKD, all-cause readmission) under the multivariable Cox model . The results showed the achieved power was 99.92%, 99.98% and 54.4% for MAKE, new-diagnosis CKD, and all-cause readmission, respectively. We have addressed this issue in the revised manuscript which is provided in the Discussion session. (Tracking change version page 8, line 292)

Point 2. Why did they not perform a multivariable analysis to delineate better the associated co-variables for AKI and AKD. I am aware that there is a significant work done on that particular subject and variables that cause AKI and AKD post cardiac surgery are well known and reported but they need to do the analysis to demonstrate that their population cohort is comparable to others reported! They can report the outcomes of such analysis as a supplementary table as they have done with other data to avoid overburdening the reader with information and risking loosing the focus of the study which is on the correlation of AKI, AKD and long term CRD!

Reply 2: We appreciate your kind reminder. We conducted a multivariable logistic regression analysis which analyzed the associated factors of AKI and AKD. We have provided it in the newly-added Supplementary Table 4 and in the second paragraph of the Discussion session. (Tracking change version page 7, line 249).

Point 3. Why did the authors/statistician utilize the Cox proportional hazards model to investigate the association between AKD incidence to long term outcomes such as MAKEs, New CKD, ESRD ect and not propensity score matching which would have been more statistically robust?

Reply 3: Thanks for your opinion. We acknowledge that propensity score matching is effective in reducing possible confounding effects. However, the number of covariates was large (>30) and the remaining sample size would be much smaller after propensity score matching, which would cause a lower statistical power. Therefore, we used multivariable regression adjustment (rather than propensity score matching) to mitigate possible confounding in this study.

Point 4. According to supplementary table 2 patients which developed AKI also had longer cardiopulmonary bypass times, clamp times and arrest times. To a cardiothoracic surgeon this is natural and expected associated variables; however I did not see this mentioned in the results text and possibly it needs to be presented and addressed for the non-surgical audience of the paper!

Reply 4: Thanks for reminding us. We supplied this important information in the first part of result session in our revised manuscript  (Tracking change version page 4, line 163)

Point 5. The discussion is well presented, poignant and succinct however I would like to see some mention about surgical causes of AKI post cardiac surgery (mentioned about performing a multivariable analysis of associated co-variables).

Reply 5: We appreciate your insightful comment. As responded in Point 2, we conducted a multivariable logistic regression analysis which analyzed the associated factors of AKI and AKD. We have provided it in the newly-added Supplementary Table 4 and in the second paragraph of the Discussion session. (Tracking change version page 7, line 249).

Point 6. Also, the authors very well mention that medications that patients receive post dissection surgery to control their blood pressure can contribute both positive but also some negatively (the authors have not taken to mention this point in their discussion) in the development of long term CRD!

Replay 6: We appreciate your insightful comment. We provided additional discussion about different impact (positive and negative) of different anti-hypertensive agents on patients outcome after AKI. (Tracking change version page 8, line 272)

Round 2

Reviewer 3 Report

Dear Editor and Authors,

It was my pleasure once again to review this manuscript titled “ Incidence and transition of acute kidney injury, acute kidney disease to chronic kidney disease after acute type A aortic dissection surgery” by Dr. Chang and colleagues.

The manuscript has undergone thorough assessment by 3 reviewers (including myself) and a number of points for improvement were raised. I was happy to see that the authors have implemented the majority of the additional analysis suggested and have addressed most of the points from the reviewers (both my own and the others) well. I am therefore now happy to recommend the publication of this work as I still consider it an “interesting study on a subject where evidence is lacking in the literature.”

I wish well to all and congratulations to the authors.

Kind regards,

Emmanouil I. Kapetanakis, MD, MSc

Cardiothoracic Surgeon